# SAGA-Supporting Social-Emotional Development in Early Childhood Education: The Development of a Mentalizing-Based Intervention

**Mirjam Kalland** [1,2,*] [ID]**, Tanja Linnavalli** [1,3] [ID] **and Malin von Koskull** [4,5]

1    Department of Education, University of Helsinki, 00014 Helsinki, Finland; tanja.linnavalli@helsinki.fi
2    Diversity, Multilingualism, and Social Justice in Education, University of Helsinki, 00014 Helsinki, Finland
3    Cicero Learning, University of Helsinki, 00014 Helsinki, Finland
4    Folkhälsans Förbund, 00250 Helsinki, Finland; malin.vonkoskull@folkhalsan.fi
5    Faculty of Medicine, University of Turku, 20014 Turku, Finland
*    Correspondence: mirjam.kalland@helsinki.fi

**Abstract:** The aim of the SAGA project is to support children's social-emotional development and teacher mentalizing by promoting regular shared story-book reading with mentalizing dialogs in early childhood education and care (ECEC) centers. The theoretical phase, the modeling phase (Phase I), and the exploratory phase (Phase II) of the SAGA intervention, as well as the research protocol for the final trial (Phase III), are described in the present article.

**Keywords:** mentalizing; social-emotional development; ECEC; shared story-book reading; intervention

## 1. Introduction

Developing an intervention model that aims to promote both children's social-emotional development within early childhood education and care (ECEC) as well as the professional development of teachers and other ECEC staff is a complex task. The SAGA intervention is based on mentalizing theory, a theoretical framework that acted as a base for developing parental groups for first-time parents in Finland [1] during the last decade. While mentalizing-based interventions have been shown to enhance parenting, especially in at-risk populations [2], the value of the theory in developing interventions in ECEC settings needs to be further explored. The need to create an intervention aiming to support social-emotional development in children, suitable for ECEC, has been widely recognized in Finland. The growing number of referrals to child psychiatry is alarming [3], and while there must be several reasons for this, it seems reasonable to conclude that children's emerging difficulties are not properly recognized and attended to in their early stages of childhood. Social-emotional and behavioral problems in development can be identified early, but they are seldom recognized by parents or child health care nurses [4]. Early aggressive behavior predicts later problems [5], and thus the notions of increased numbers of children with conduct problems or other problems related to social-emotional development in the early years are worrying and need attention. A summary of support for children's social and emotional development in ECEC in Finland [6] concluded that there is an urgent need for feasible intervention tools in ECEC centers. A meta-analysis has shown that interventions tend to be more influential when they are led by researchers or other experts, rather than ECEC teachers [7]. This calls for the development of practices based on a strong theoretical foundation that are easy to integrate with the daily routines of ECEC centers, are targeted at all children, and are readily adopted by all the staff. As children's mentalizing abilities have been positively linked to their social skills [8], a carefully developed and conducted intervention based on mentalization theory has the potential to become an influential tool in supporting social-emotional development. In addition, ECEC

interventions using discussions about mental states have been successful in supporting mentalizing [9]. In this paper, the various steps in the development and the theoretical framework of the SAGA intervention are described in detail.

### 1.1. A Stepwise Framework for Designing and Evaluating Interventions

The Medical Research Council (MRC) in the United Kingdom has developed a stepwise framework for designing and evaluating complex interventions [10]. According to the MRC framework, the first step is pre-clinical or theoretical, and answers the question of why this intervention is important and clarifies the mechanisms by which the intervention ought to work in theory. The second step (Phase I) answers the question of how the intervention will be implemented, including practicalities and the concrete structure of the intervention. Phase II involves conducting an exploratory trial, with opportunities to refine both the intervention and the research protocol. In Phase III, a definitive randomized controlled trial is conducted. The final phase after the trial is implementation and dissemination.

In a revised version of the MRC framework published in 2006 [11], greater attention was given to the early phase of piloting and development work, as well as to the need for adapting interventions to the local context. In addition, a consideration of alternatives to randomized trials was suggested. Although the MRC framework was developed with health service delivery in mind, the framework may be used in social policy and education when the expected outcome is related to health, wellbeing, and development [10]. The theoretical (preclinical) phase, the modeling phase (Phase I), and the exploratory phase (Phase II) of the SAGA intervention, as well as the planned final trial and its research protocol, are described in the present article.

### 1.2. Rationale and Theory behind the SAGA Intervention

As mentioned above, the importance of supporting children's social-emotional development from early years and beyond is widely recognized and has been linked with later capacity for empathy and prosocial behavior as well as with school readiness and later academic achievement [12,13]. The formative years between 3 and 6 present a unique opportunity to support children's social and emotional development, as the development of language and other skills is rapid, and children are receptive to guidance and support [7]. In addition to parents, emotionally supportive teachers are potentially important for supporting children's social-emotional development. In terms of designing interventions, it is impossible to reach every parent and change potentially harmful patterns of parenting. However, it is possible and necessary to constantly improve the quality of ECEC, which reaches nearly all children when they get to the age of five or six years at the latest. In the EU, the explicit goal is to increase participation in ECEC among 4-year-olds and older children [14]. Thus, it is important to focus on supporting children's social-emotional development within the field of early childhood education and care.

### 1.3. Caregiver Mentalizing and Child Development

Parental mentalizing is the "art of reflecting on the mind behind a child's behavior" [1]. What are the needs, intentions, emotions, or thoughts behind the explicit behavior of the child? What is the child trying to communicate, and how can a parent best meet the needs of the child? Parental mentalizing refers to the parent's capacity to think about and understand the child's behavior in terms of mental states, such as emotions and intentions. Research in parenting has revealed that the caregiver's capacity to mentalize is associated with more sensitive interaction with the child and more secure child attachment compared to caregivers showing lower mentalizing skills [15,16]. Furthermore, a parent's capacity to mentalize is important for promoting the young child's growing capacity for emotional regulation, self-organization, and social competence, including resilience to adversity. Gottman and colleagues suggest that optimal parenting in terms of preschool children's psychosocial adjustment is based on an "emotion-coaching philosophy" that

includes parental awareness and validation of low-intensity emotions in themselves and in their children, as well as assisting the child in verbally labeling his/her emotions [17].

Mentalizing is the core concept used as a base for SAGA intervention. However, parental reflective functioning is sometimes referred to as (1) the operationalization of the mental processes that underpin the capacity to mentalize, as well as (2) the observed manifestation of the capacity to mentalize when interacting with the child, or (3) simply as parallel concepts [18]. Others define mentalizing as having three components: parental mind-mindedness, parental insightfulness, and parental reflective functioning [19]. However, while the relationship between mentalizing and mind-mindedness remains open to debate [20], parental reflective functioning provides an empirically grounded framework for understanding the ability to mentalize [21]. Parental reflective functioning can be assessed through interviews [16] or questionnaires [22]. Parental or caregiver mentalizing is important for child social-emotional development, and low parental reflective functioning has been negatively related to child social-emotional development [18]. According to a review, studies so far have given support to the value of enhancing reflective functioning in parenting intervention programs [2].

The SAGA intervention is based on mentalizing theory, recognizing the literature using both the concepts of mentalizing and reflective functioning. As there is evidence that small children may form attachment relationships with teachers [23], it seems fruitful to consider teacher mentalizing and reflective functioning in the context of forming emotionally supportive classrooms in ECEC. While some teachers are able to form supportive and warm relationships with children with disruptive or challenging behavior, other teachers may become overwhelmed by feelings of anger or helplessness. Jennings and Greenberg [24] have suggested that teachers' social-emotional competence is an important construct that is fundamental to teachers' ability to provide an emotionally supportive classroom and foster children's social-emotional competence. However, the social-emotional competence of teachers varies, and it is therefore important to create a model of interaction that improves the competence of the teachers and other staff in the ECEC units and provides the children with regularly occurring emotionally supportive moments.

A pilot study evaluated teacher mentalizing and reflective functioning using the Parent Development Interview Revised for Teachers (PDI-R/T) in a pre-school setting [25]. Teachers rated as highly reflective by a self-report questionnaire also gave significantly more self-reported examples of using behaviors known to foster social-emotional skills than teachers rated as showing moderate or low reflective functioning. These data, although preliminary, indicate that reflective functioning may underlie the behaviors that teachers use when interacting with children in a developmentally supportive way. One conclusion is that enhancing reflective functioning in teachers can help them build coherent and complete mental representations of particularly challenging children and secure relationships with them, thus enhancing children's emotional self-regulation and social competence. Emotionally supportive and secure classrooms may help children pay attention to adults and rely on them as a reliable source of information, as indicated by the notions of epistemic trust in securely attached children [26–28].

All in all, a mentalizing teacher could be described as utilizing the reflective parenting stance and adding to the description of it [29], namely:

- a benign interest in the mind of the children, and emotional availability to help the children make sense of their own reactions as well as those of others, including helping the children to find the words to express their feelings
- the "not-knowing-stance": the capacity for not jumping to fast conclusions, and keeping one's mind open to alternative perspectives
- a capacity to look past the behavior to determine what is going on in the mind of a particular child, and what he or she is trying to communicate
- the capacity to play, joke, and imagine with the children
- the motivation to see the perspective of individual children and an awareness of it that it might differ from one's own and from that of other children

- an ability to have a sense of one's own thoughts and feelings and to regulate and modulate one's own behavior when interacting with the children
- an appreciation that one's own feelings and moods might have an impact on an individual child, as well as on the group

*1.4. Shared Story Book Reading as a Safe Space for Exploring Emotions*

Research has shown that social-emotional development is closely associated with language development (for reviews, see [30,31]). In particular, there is evidence that the development of children's mentalizing capacity and their social-emotional development is supported by the caregiver's versatile use of language and references to mental states [32–38]. Previously, shared story-book reading (SSBR) had been found to support language development [39–41] but there is also preliminary evidence for its usability in supporting social-emotional development in small children [42]. In SAGA, SSBR is used to support children's language development as well as social-emotional development, with the emphasis on supporting children's interest and understanding of the nature of mental states underlying explicit behavior, starting from supporting the ability to recognize and name feelings and emotions.

As story characters allow both the teacher and the child to explore feelings, thoughts, and intentions at a distance from real social situations, shared story-book reading may be a safer context for promoting children's social-emotional competence than other interactions. Children may have difficulties in containing their feelings and reflecting on inner mental states when they try to deal with real situations, such as conflicts with peers. In addition, children who have been exposed to trauma will be likely to react with strong feelings and disruptive behavior even in normative stressful situations, such as separations from caregivers. Mentalizing is more difficult while distressed [43] and in relation to trauma [44] compared to situations lacking these factors. Thus, the stories can be seen as a safe space in which children with the support of empathic adults will be able to think about and find words for feelings and other mental states, while feeling emotionally protected. As the stories are interesting but not overwhelming, this safe space can provide an opportunity for the child to reach the optimal zone of arousal in the "window of tolerance" [45]. As opposed to zones of hyper- and hypo-arousal, the optimal zone of arousal facilitates thinking, feeling, and self-regulation. This view is supported by a meta-analysis examining the relations between parents' mental-state talk and children's social understanding by Tompkins et al. [46] that found books to be a useful tool for discussing mental and emotional states during the preschool years. In the SAGA intervention, SSBR is regarded as a useful means for facilitating mentalizing dialogs and promoting children's social-emotional development, as well as the reflective functioning of teachers. The children and the teacher can reflect together on the experiences, motives, and emotions of the characters in the stories, without pre-given interpretations or "right" answers. In SAGA, we find it important for children to develop a truly representational understanding of the mind by exploring opportunities for understanding and interpretation, rather than simply mimicking the language they hear from caregivers or peers [47].

*1.5. Conclusions from the Pre-Clinical Phase: SAGA Intervention and Hypotheses*

Based on the theoretical perspectives above, a mentalizing-based intervention using SSBR was developed. The aim of the intervention is to support the social-emotional development of 3–5-year-old children and teacher/other staff mentalizing skills in ECEC. Language development and social-emotional development are linked, as was shown by the first, interrupted pilot study [48], and the aim of SAGA intervention is to support children's deeper understanding of words with reference to mental states and emotions, as well as their social-emotional development. The hypotheses are that SSBR with mentalizing dialogs will:

- Enhance child social-emotional development and prosocial behavior
- Support teacher/caregiver mentalizing

By promoting children's prosocial behavior and teacher/caregiver mentalizing, SAGA will also enhance a positive group/classroom climate in ECEC.

*1.6. Phase I: The Structure of the SAGA Intervention*

The SAGA intervention aims to support children's social-emotional development and teacher mentalizing by promoting SSBR among 3–5-year-old children. The teachers are encouraged to support the children's interest in the motives, emotions, and thoughts of the characters in the stories by having mentalizing inner-state conversations with the children after or during the shared moments of reading. Mentalizing dialogs in SAGA are defined as dialogs that enhance children's interest in and understanding of emotions, thoughts, and intentions in themselves and in others, especially in relation to mental states underlying behavior. In addition, SAGA supports ECEC teachers' mentalizing and reflective functioning in the interaction with the children. Storybooks and fairy tales produced for children have often been shown to contain rich references about the mental states of the figures in the story [49]. To achieve the aims of SAGA, the stories used in the intervention have been carefully selected, to provide opportunities to reflect on the inner mental states behind overt behavior. For each story, a dialog card is created with suggestions for reflections about what happens in the story, and what might go on in the minds of the characters in the story.

The mentalizing-promoting stance of the teacher during the SSBR sessions includes: (1) Taking time to identify feelings, motives, or thoughts; (2) maintaining a (humble) stance of not knowing for sure; (3) accepting alternative perspectives and suggestions from the children (no right or wrong answers); (4) encouraging the children to explain things in their own way ("tell me what happened here? What do you think about that?"); and (5) accepting that some things in the story may remain unclear (these points are adapted from the description of a therapist's mentalizing therapeutic stance in [50]. In addition, the teachers are encouraged to start with funny questions, such as "Do you have a dinosaur at home?", to maintain an atmosphere of humor and an acceptance of playfulness. The SAGA sessions also make it possible to stop for a moment and explore together the deeper meanings of common words, such as "being in a hurry"—what does it mean? How does it feel when an adult is hurrying? What does a hurrying adult look like?

Some of the stories might evoke strong feelings in both the children and the teachers, and teachers are encouraged to attend to those feelings with acceptance and empathy.

Mentalization-based reflections in the SAGA model can be categorized into four levels of complexity (see Figure 1). These levels are adapted for small children and inspired by the various levels of child reflective functioning developed in the Child and Adolescent Reflective Functioning Scale (CRFS), used to code the Child Attachment Interview (CAI) [51,52].

The first level is about recognizing and naming mental states, such as feelings and emotions. The teacher may use emotion cards (for example, pictures of a teddy bear with different emotions) to help children think about and recognize emotions. On the second level, the teacher explores how explicitly observable behavior is linked with possible inner states of the children; for example, a child kicking his/her bike, in anger or disappointment as an emotion underlying the explicit behavior. At this level, the teacher/caregiver may ask "What do you think, what is going on in the mind of "Lotta" (character in a story) when she kicks her bike?" The third level includes reflections on how it is possible to impact each other's emotions and mental states, such as how it might feel from the child's perspective when the parent comforts her or him. A question at this level may relate to how it feels when a mother hugs and comforts the character in the story. This level also includes reflections on how it can be helpful for somebody when a third person (e.g., an adult) helps to solve a conflict (e.g., children quarreling). The fourth level is the most complex one, manifesting reflections on complex and conflicting emotions, such as being afraid and curious at the same time. This level also includes reflections on characters in the stories that might want to hide emotions or intentions, or on feelings changing over time. On this

level, the teacher/caregiver can also discuss self-regulation with the children—one might need to learn to wait, or not do what one would like to, or find new ways of expressing feelings and emotions.

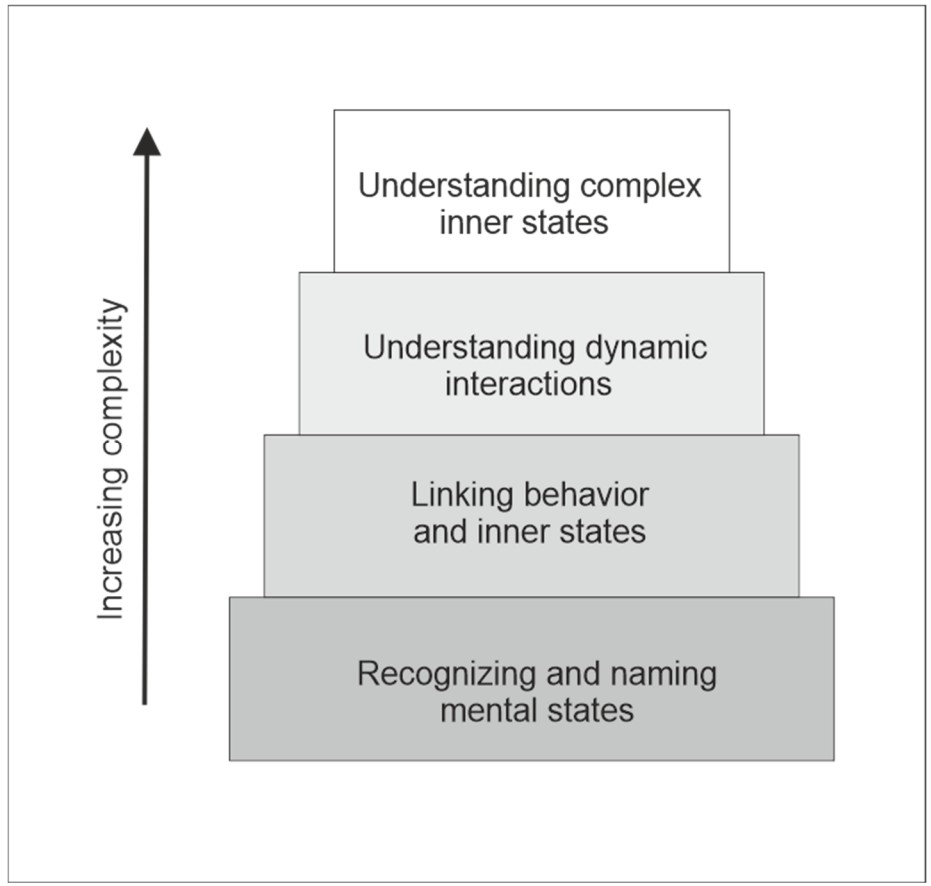

**Figure 1.** A tentative model of supporting child mentalizing.

The staff in the ECEC centers are provided with four hours of training (two afternoons) in mentalizing theory and the SAGA model, plus two workshops during the intervention. SAGA training is targeted at all ECEC center staff, not only at teachers. During the training, the staff are presented with theoretical perspectives, and stories and fairy tales are used for mentalizing together. The participants and educators think about the motives, affects/emotions, and intentions behind the explicit behavior of the characters in the stories. Alternative ways of handling the episodes in the stories are also discussed. Instead of asking questions about mere happenings or cognitive processes behind the actions, the emphasis of the dialogs is on emotions, affects, desires, and intentions. After the training, staff are offered two workshops during the 12-week intervention. During the workshops, staff members reflect on their experiences of the intervention, and create new dialog cards of self-brought stories or fairy tales with the educators. An important agenda of the training is also enhancing mentalizing and reflection of the teachers so that they become more aware of the importance of the not-knowing stance, understanding the opacity of mental states, and of being genuinely interested in and curious about the inner world of the children.

The SSBR-based SAGA model is composed of three weekly SAGA sessions for each child. The staff at the ECEC centers are asked to arrange these sessions in groups of 5–6 children, each session lasting for 15–20 min. The formation of the groups is decided by each unit, but the groups should be stable (i.e., the children are in the same group the whole intervention). During the training, the formation of the groups is discussed, for example, the possible advantage of counterbalancing more expressive children with children that are usually more silent during group sessions. The staff are also encouraged

during the training to make efforts to provide space for less expressive children and to gently ask all children to express their thoughts. The teacher can choose to read the same story several times or a new story each time, depending on the needs and interests of the group. In general, the mentalizing dialogs take place after the story reading, but sometimes the children may ask questions in the middle of the story. If that happens, the teacher is advised to use the opportunity to discuss with the children for a short while, and then go on with the reading. If the story is long, the teacher can interrupt the story and continue with it the next day.

*1.7. Example from the Story "Will You Be My Friend"*

"Will you be my Friend" by Sam McBratney is a tale about friendship. One day, Big Nutbrown Hare is busy, but Little Nutbrown Hare wants to play. Little Nutbrown asks if he can go off exploring on his own. During his trip, he discovers his reflection in a puddle and spots his shadow. Later, on Cloudy Mountain he finds a little snow-white hare called Tipps, who becomes his friend, but they lose each other when playing hide and seek. Finally, Tipps shows up when Little Nutbrown has returned home. After the story, the teacher/caregiver reflects with the children on:

1. What does it mean to be busy? What do you think, how does it feel for Little Nutbrown when Big Nutbrown is busy?
2. How do you think Little Nutbrown felt when he/she met Tipps and Tipps asked if they could be friends?
3. How can you see when somebody is having fun? Do you think it is more fun when Little Nutbrown has fun alone, or when Little Nutbrown and Tipps have fun together? Tell me more about that!
4. What did Little Nutbrown feel when Tipps shows up again after they have lost each other? What do you think Tipps felt? How can you show somebody that you are happy to see him/her/it again?

These questions are examples, and the teacher/caregiver may also discuss other topics, relevant to the children. Furthermore, the children may generate questions and answers that the group will reflect on together. During the pilot phase of the study, some SAGA sessions were videotaped. According to these videos, the teachers easily stray into asking dichotomic questions that can be answered with "yes" or "no" by the children. Thus, in the dialog cards, the importance of asking open questions is emphasized.

*1.8. Considerations and Justifications for Intervention Content and Structure*

Child social-emotional development is often described as emerging skills or competence that can be learned within educational contexts [53]. However, in SAGA, the emphasis is on supporting social-emotional development, rather than on teaching social-emotional skills. If disconnected from development, teaching specific skills may happen during isolated occasions having no connection to the daily practice in the ECEC centers. In the worst-case scenario, this may lead to situations in which kindness is taught but children are treated with harshness. In SAGA, the importance of not seeing the child as being less skilled, or solely as having problem behaviors that should be changed, is emphasized. A double agenda is therefore considered to be important in SAGA: in addition to enhancing mutual reflection with the children as described above, we aim to support the mentalizing skills of the staff, including all staff and not only the teachers at the unit. To feel understood and valued, the child needs a warm and supportive caregiving environment with caregivers who are interested in what is going on in the mind of the child and aim to help him/her develop a true understanding of his/her own and other's mind. The dialogs contain horizontal (symmetric) qualities, which have been associated with secure attachment and thus may enhance a secure teacher–child relationship [54]. In addition, horizontal qualities in interaction may improve children's peer skills [55] as well as add to children's sense of participation. While a "child-centered approach" is characterized by asymmetrical, vertical interaction [55], horizontal interaction qualities could be described

as enhancing a child's perspective and child participation, thus implementing one of the basic principles of the United Nations Convention on the Right of the Child [56].

Instead of focusing only on "kids that are in need of special support", including all children in the intervention is important in the SAGA intervention. This is in line with the Salamanca Statement of inclusion [57]. Inclusive groups may help children to learn from each other in a group, and with the gentle support of the teacher/caregiver, learn to function better as a group. Participating in group activities, such as SAGA sessions, might also enhance children's ability to listen to, support, and comfort each other, thus further adding to the experience of being heard and understood. If the intervention is successful, the improved mentalizing capacities of the staff and the improved prosocial behavior of the children will affect the group and classroom climate overall, in all activities. Reflecting together in a group might help children to learn from each other, as some children will be better prepared to reflect on feelings and emotions than others.

Finally, the SAGA intervention may be effortlessly integrated with the daily activities in ECEC and is cost-free (except for training in the future) as the fairy tales and stories can be chosen from among the stories that are already in use in the ECEC unit or can be borrowed from a library. The SAGA intervention may also be used in parallel with other interventions, such as "Second Step" [58].

### 1.9. Phase II: The Modelling Phase and First Implementation

Originally, the SAGA intervention was to be pilot tested in March 2020 in Helsinki metropolitan area, Finland. However, the COVID-19 pandemic interrupted the intervention and the research. The first pilot test resulted in two papers, one estimating the impact of the pandemic on the social-emotional wellbeing of the children and families [59] and one evaluating the rationale for the study [48].

In January 2021, the researchers involved in the SAGA project were invited to participate in an intervention study coordinated by another project, CREDU [60] comparing two interventions and their effects on children's social-emotional development. This follow-up study did not allow for using the originally intended tests or interviewing the children, as the researchers were not permitted to enter the ECEC centers. Moreover, the staff training was conducted remotely.

Based on the experiences from the modeling phase, the structure of intervention, the training, and the research protocol were modified. Organizing the training remotely, as well as engaging two of the staff to be responsible for the intervention and gathering research material was deemed possible and even recommended, as it engaged these staff more in the interventions and the research. In addition, a meeting with the parents before the start of the intervention was added to the intervention protocol. During this meeting, the parents were informed about SAGA and the importance of story-book reading for supporting child social-emotional and language development.

### 1.10. Phase III: Evaluation of the SAGA Intervention

The next stage of SAGA is a case-control study using the revised research protocol. SAGA research has been launched in the Helsinki metropolitan area and in western Finland. ECEC centers interested in the model were assigned to either intervention or active waiting-list control (diary of reading activities, without intervention-specific dialogs).

The hypotheses of SAGA are that shared story-book reading with mentalizing dialogs will:

- enhance emotional understanding
- enhance child social-emotional development in three areas, i.e., internalizing, externalizing, and prosocial behavior
- support teacher mentalizing
- enhance positive group/classroom climate in ECEC units

In addition to the hypothesis, we explored whether SAGA would have an impact on the expressive language development of the children and if SAGA could have an impact on the wellbeing of children, as reported by the children themselves.

## 2. Methods

The duration of the intervention is 12 weeks. To evaluate possible short-term effects of the SAGA intervention, children are tested by research assistants and evaluated by the teachers in the pre- and post-intervention stage (see Figure 2 and Table 1). In addition, the staff at the ECEC units answer questionnaires pre- and post-intervention. To assess intervention model fidelity, the staff keep a diary of SAGA sessions, and two sessions in each unit are video recorded during the intervention and analyzed using conversation analysis. Finally, the staff will take part in a focus group interview regarding the experiences of the intervention. To evaluate possible long-term effects of the intervention, the children will be tested and evaluated again one-year post-intervention with a shorter protocol. The research protocol for SAGA is displayed in Figure 2.

**Figure 2.** The research protocol for SAGA. * Research assistants testing the children. ** Teachers filling in the questionnaires.

**Table 1.** Study protocol. RA = research assistant. ECEC = early childhood education and care.

| Test/Assessments | Target Group (Who Tests/Fills) | Timepoint |
|---|---|---|
| Word Generation | Children (RA) | Before and after intervention |
| Teddy Bear Test | Children (RA) | Before and after intervention |
| Kiddy-KINDL® assessment | Children (RA) | Before and after intervention |
| Strengths and Difficulties Questionnaire (SDQ) | Children (staff) | Before and after intervention |
| MASK (6 items) | Children (staff) | Before and after intervention |
| MentS | ECEC staff (staff) | Before and after intervention |
| Group assessment | Children (staff) | Before and after intervention |
| Video recordings of SAGA reading sessions | Children and staff (RA) | During intervention |
| Diary of SAGA reading sessions | ECEC staff | During intervention |
| Focus group interviews | ECEC staff (RA) | After intervention |

## 2.1. Participants

Approximately 200 children from 9 ECEC centers were recruited for the study. The participants were aged 3–5-year-old kindergarten students and the ECEC centers were in the Helsinki metropolitan area and in western Finland. The ECEC centers were divided into SAGA kindergartens and control kindergartens. The research plan was approved by the University of Helsinki Ethical Review Board in the Humanities and Social and Behavioural Sciences, in Helsinki, Finland, and was carried out in accordance with the committee's guidelines and regulations, as well as with those of the Helsinki Declaration.

## 2.2. Assessments and Questionnaires

ECEC teachers filled in the Strengths and Difficulties Questionnaire (SDQ) [61] and selected questions from the MASK test [62] for each child. They also assessed group climate with a questionnaire developed for the SAGA intervention. SAGA project's research assistants administered part of the Word Generation test (NEPSY II) [63] and a Teddy Bear test (TBT) constructed and piloted for SAGA research [48], for all participating children. Additionally, they interviewed children using a revision of the KiddyKindl protocol [64]. Teachers/other staff were to complete a mentalizing questionnaire, and caregivers provided information about children's language background, socio-economic status, extra-curricular activities, and amount of reading for children. Part of the SAGA sessions was videotaped for conversation analysis, and the ECEC staff filled in a diary of the SAGA sessions. Two staff members at each ECEC center were assigned and trained for taking charge of implementing the intervention and gathering research data. All the assessments and tests are listed in Table 1.

## 2.3. Word Generation

The Word Generation test measures expressive verbal development. The child is asked to say aloud as many examples of specific semantic and initial letter categories as possible in one minute. Due to the young age of the children, only two categories were used in SAGA, namely animals, and foods and beverages.

## 2.4. Teddy Bear Test

The Teddy Bear test (TBT) was developed for the SAGA project to assess children's capacity to understand, recognize, and name emotions related to a story, with the focus of assessing children's social-emotional understanding and capacity to empathize with the story characters. The test was inspired by the story stem assessment profile [65] which assesses a child's expectations of family relationships indirectly and has been used previously by the first author (a child psychotherapist).

In the test, the child is told that her or his task is to help Teddy to find his emotions, and that a feeling could be that one is happy, sad, or scared. Four cards with a teddy bear drawn on them are laid on the table. Each card shows an expression and a posture reflecting different emotions. The experimenter tells the child a short story excerpt about the teddy bear (that Teddy could not find his cuddly toy) and the child is asked to point at a card that reflects the teddy's emotion. The right option for each story is listed in the test protocol. Choosing an appropriate card gives one point and being able to name the emotion in the card gives two more points. If the child cannot name the emotion but uses an appropriate verb (e.g., the child answers "Teddy laughs" instead of "Teddy is happy"), he/she is given one instead of two points. Thus, each question is scored with $0-3$ points. The maximum scores are 30. If the child answers "I don't know" the research assistant is instructed to say, "It's ok, it is not so easy to know what Teddy might feel". The TBT has been used previously [48] and has been revised since then. With 90 participants, TBT showed small to moderate correlations with Theory of Mind, Affect Recognition and Word Generation (NEPSY II) [63], as well as with Vocabulary (WPPSI IV) [66] subtests [48]. The test has been created for evaluating development in recognizing and naming emotions.

### 2.5. Strengths and Difficulties Questionnaire

The Strengths and Difficulties Questionnaire (SDQ) assesses the behavior of children and adolescents [61]. The questionnaire is validated for assessing social-emotional development [67,68] and has been developed for use by clinicians, researchers, and educationalists. SDQ may be filled in by teachers, parents, and by older children and adolescents. Furthermore, it is easy to administer, and available in many languages. The 25 items are divided into five scales, which are then further remodeled to three factors: (1) internal symptoms (emotional symptoms and peer relationship problems), (2) external problems (hyperactivity/inattention and conduct problems), and (3) prosocial behavior. Each of the five scales comprises five statements regarding the child in question, such as "is kind to younger children", "often lies or cheats", "generally liked by other children" and "often loses temper". Each item is assessed with not true = 0, somewhat true = 1, or certainly true = 2.

### 2.6. MASK

MASK is a multiple assessment of the social competence questionnaire [62]. The questionnaire was originally created for assessing elementary school students' social competence and may be filled in by teachers, students, or students' peers. In SAGA, six items from MASK are adopted and filled by the teachers for each child. The items deal with children's behavior with their peers (e.g., "Eagerly participates in the group's activities"), and are answered on a four-point scale: 1 = the child never behaves in this way, 2 = the child rarely behaves in this way, 3 = the child often behaves in this way, 4 = the child behaves very often in this way.

### 2.7. KiddyKindl

The KINDL® questionnaire [64] is an instrument for measuring children's and adolescents' quality of life. It consists of several questionnaires for different age groups and may be filled in by children or adolescents, and their parents and teachers. The Kiddy-KINDL® questionnaire for 4–6-year-olds is used in SAGA to obtain the children's perspective on their own well-being. For such small children, the questionnaire is filled in by interviewing them. The interview includes two items for each of the six factors (physical well-being, emotional well-being, self-esteem, family, friends, and preschool). The interviewer asks the child if he/she "has fun and laughs a lot", and the child chooses between the alternatives of "rarely", "sometimes", and "very often". Originally, Kiddy-KINDL® used the expression "past week" but this seemed not to be clear for small children (as tested previously), and the sentence was rephrased. Moreover, the answer "never" was changed into "rarely", as it was in a pilot testing phase considered a more culturally appropriate expression. The KINDL® questionnaire has been deemed to have good reliability and validity among school-aged children and adolescents (e.g., [69,70], but there seems to be less support for the reliability of the self-assessment of preschool-aged children [71]. However, in SAGA, Kiddy-KINDL® is used for the purpose of enhancing child participation in evaluating changes in his/her wellbeing and peer relations.

### 2.8. MentS: Teacher/Caregiver Mentalizing

The ability to mentalize and changes in it are assessed using a teacher and professional caregiver version of the MentS self-reporting questionnaire [72]. MentS was originally tested with employed adults and university students (non-clinical sample) as well as with a sample of persons with borderline personality disorder (clinical sample), and the internal consistency and reliability were higher for the non-clinical sample. MentS contains 28 items assessing the ability to mentalize (whole scale) and may be divided into three underlining dimensions, i.e., self-related mentalization, other-related mentalization, and motivation to mentalize. In SAGA, the MentS assessment has been translated to Swedish and adapted to fit teachers and professional caregivers. For example, the item "I often think about other people and their behavior" is transformed to "I often think about the children in my group and their behavior". The items are answered on a five-

point scale: 1 = completely incorrect, 2 = mostly incorrect, 3 = both correct and incorrect, 4 = mostly correct, 5 = completely correct.

### 2.9. Group Climate

A staff member filled in a questionnaire developed for SAGA, probing the empathy, willingness to help, self-regulation, and cooperative behavior in the ECEC group. The 12 items include, e.g., statements "children comfort each other" or "children try to follow the rules", and they are answered on a five-point scale: 1 = no one in the group, 2 = some children in the group, 3 = half of the group, 4 = most children in the group, 5 = all children in the group.

### 2.10. Diary

The person in charge of the SAGA sessions filled in a diary after each session. The date, time, and duration of each session are asked for, as well as the name of the story and whether a previously created dialog card was used to support the conversation. In addition, the number of the children and their names (if they are participating in the study) are collected. Most importantly, comments on the sessions and examples of the children's questions and attitudes, are written down. The control ECEC centers fill in a diary about their reading sessions in a similar way to control for the effects of reading activities per se.

### 2.11. Focus Group Interviews

The focus group interview is considered an ideal venue to generate new knowledge for experiences that might be difficult to assess otherwise (see [73]). In the current study, the focus group interviews are used to estimate the feasibility of the SAGA model as experienced by the users in the ECEC unites, as well as for exploring together with the users' experiences that the researchers did not think about evaluating, including possible negative experiences of the model for the staff or for the children.

## 3. Discussion and Limitations

The rationale for SAGA is based both on the opportunity to support social-emotional development through shared story-book reading with mentalizing dialogs, as well as on the observed links between social-emotional and language development. The first results from the interrupted baseline research confirm the rationale for SAGA about language development [48]. We found the expected correlations between language development and different dimensions of social-emotional development. This supports the rationale behind the SAGA project that aims to enhance both children's language development and their social-emotional understanding to support social-emotional development. The links between mentalizing dialogs in ECEC and children's social-emotional development will be studied in the trial.

A general challenge related to evaluating an intervention is the stability and loyalty of the model: We cannot ensure that all the SAGA sessions are conducted according to the training. However, we have videotaped SAGA moments twice during the intervention period to be able to observe the quality of the dialogs between children and the staff. In addition, the staff have kept a diary of the SAGA moments. The feasibility of the SAGA model will be evaluated by conducting Focus Group Interviews with involved staff after the intervention.

Another challenge related to dissemination is related to the continuity and dissemination of the model: If the evidence supporting the intervention is sufficiently strong, how do we ensure that the intervention model will remain alive and well and in active use? Moreover, how can we spread the model, at least nationally? A long-enough intervention (12 weeks) combined with a follow-up study and support for staff in form of workshops might be helpful in the process of integrating the method with daily ECEC activities. We will also follow the ECEC units after the intervention to find out if they have continued to have SAGA sessions. Furthermore, the participating children will be measured later with

partly the same assessments and questionnaires to study the possible long-term effects of the intervention. In the case of SAGA, the model can also become integrated with the academic program for student teachers at the University of Finland and available for further education for teachers and other staff working within ECEC through Open University as a part of in-service training.

## 4. Conclusions

Without results of the effects of the SAGA model, it is difficult to make final conclusions of the impact or importance of the SAGA intervention. However, the emphasis on supporting children's social-emotional development as early as possible is currently considered important, and several other research teams have recently presented interesting approaches and results. For example, Brazzeli and colleagues [74], as well as Ornaghi [75] have presented results based on approaches very similar to SAGA, albeit not explicitly based on mentalizing theory. In addition, the focus is on feasibility and low-cost interventions [76], since interventions that require long training and/or financial resources are not likely to become adopted in ECEC settings with often high rates of staff turnover and shortage of resources. In the case SAGA will show short and/or long-term effects on child development, SAGA is easily replicable and will ultimately be internationally available for anyone interested.

**Author Contributions:** Conceptualization, M.K. and M.v.K.; methodology, M.K. and T.L.; writing—original draft preparation, M.K. and T.L.; writing—review and editing, M.K., T.L. and M.v.K.; visualization, M.K., M.v.K. and T.L.; project administration, M.K.; funding acquisition, M.K. All authors have read and agreed to the published version of the manuscript.

**Funding:** The research has been funded by The Swedish Cultural Foundation in Finland.

**Institutional Review Board Statement:** The research plan was approved by the University of Helsinki Ethical Review Board in the Humanities and Social and Behavioural Sciences, in Helsinki, Finland, and was carried out in accordance with the committee's guidelines and regulations, as well as with those of the Helsinki Declaration.

**Informed Consent Statement:** Informed consent was obtained from all subjects involved in the study.

**Data Availability Statement:** Not Applicable.

**Conflicts of Interest:** The authors declare no conflict of interest.

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
