# Peer review of "SAGA-Supporting Social-Emotional Development in Early Childhood Education: The Development of a Mentalizing-Based Intervention"

_education, doi:10.3390/educsci12060409_

Round 1

Reviewer 1 Report

Comments to ms SAGA– supporting social-emotional development in early childhood education. The development of a mentalizing-based intervention.

This is a well-written paper and seems an interesting attempt to study the effects of a story reading intervention on socio-emotional development and vocabulary development.

As a study protocol I am positive for potential publication. However in this case I suggest to reconsider all those parts that present preliminary findings, given that the presentation of them is naïve and not according to typical methodology of presenting research findings.

Author Response

Thank you so much for using your valuable time for improving our paper, and for your encouraging comments. We have now revised the manuscript according to your suggestions, and hope we have answered all your comments appropriately. The revised texts are marked with green color in the MS.

Regarding to your detailed assessments and comments:

  • as none of the authors is a native English speaker, we will use editing services of the journal after the possible acceptance of the manuscript
  • we have now included more current empirical research in the MS (especially in the Conclusions section)
  • we have now clarified the research design, page 10 and 11, and added a new Table, page 14
  • we have eliminated references to preliminary findings, page 6 rows 235-238, page 8 rows 371-373 and page 14, rows 554-556. The pilot findings will be presented in a forthcoming paper
  • we have now developed the Conclusions session

Reviewer 2 Report

This is an important and valuable project. In Europe there are some research teams working in the same line and it can be worthwhile to check and add references.

Italy: https://www.labpse.it/en/pubblicazioni-scientifiche 

Spain: https://pensandoemocionesatencionplena.wordpress.com/como-lo-hacemos/investigacion/publicaciones/ 

Maybe it would be interesting to know how to choose children to compose the small groups. It is important to counterbalance children more expressive with other that are not so expressive, children who are not so language skilled with skilled language children.

Furthermore, I really appreciate your effots working in this area and highly recommend the article's publication. Congratulations.

Author Response

Thank you so much for using your valuable time for improving our paper, and for your encouraging comments. We have now revised the manuscript according to the suggestions, and hope we have answered all your comments well enough. The revised texts are marked with green color in the MS.

Regarding to your detailed assessments and comments:

  • we have added current and relevant references to ongoing research (especially in the Conclusions section)
  • we have added current instructions on how to compose the child groups, page 7 and 8.
  • we have now developed the conclusions section

Round 2

Reviewer 1 Report

This manuscript can be pubished now as a research proposal format.